# Neural Collapse meets Differential Privacy: Curious behaviors of NoisySGD with Near-Perfect Representation Learning

## Abstract

Recent study by De et al. (2022) reports that large-scale representation learning via pre-training on a gigantic dataset significantly enhances differentially private learning in downstream tasks. By training on Google's proprietary JFT dataset, they achieved a remarkable 83% Top 1 accuracy on ImageNet with strong privacy parameters, despite the high dimensionality of the feature space. While the exact behaviors of NoisySGD on these problems remain intractable to analyze theoretically, we consider an idealized setting of a layer-peeled model for representation learning, which results in interesting phenomena of the learned features known as neural collapse. In such a setting, we observed several curious behaviors of NoisySGD. Specifically, we find that under perfect neural collapse, the misclassification error is unaffected by the dimension of the private training set for any learning rate. This finding is consistent even with class imbalance and remains unaffected by the nature of the loss functions. Nevertheless, a dimension dependency emerges when introducing minor perturbations in either the feature or model space. To mitigate this non-robustness under perturbation, we suggest several strategies, such as pre-processing features or employing dimension reduction methods.

## 1 Introduction

Recently, privately fine-tuning a publicly pre-trained model with differential privacy has become the workhorse of private deep learning. For example, De et al. (2022) demonstrates that fine-tuning the last-layer of an ImageNet pre-trained Wide-ResNet achieves an accuracy of $95.4\%$ on CIFAR-10 with ($\epsilon = 2.0, \delta = 10^{-5}$)-DP, surpassing the $67.0\%$ accuracy from private training from scratch with a three-layer convolutional neural network (Abadi et al., 2016). Additionally, Li et al. (2021); Yu et al. (2021) show that pre-trained BERT (Devlin et al., 2018) and GPT-2 (Radford et al., 2018) models achieve near no-privacy utility trade-off when fine-tuned for sentence classification and generation tasks.

However, the empirical success of private fine-tuning pre-trained large models appears to defy the curse of dimensionality in private models — noisy stochastic gradient descent (NoisySGD) requires adding noise scaled to $\sqrt{d}$ to each coordinate of the gradient in a model with $d$ parameters, rendering it infeasible for large models with millions of parameters. This suggests that the benefits of pre-training may help mitigate the dimension dependency in NoisySGD. A recent work (Li et al., 2022) makes a first attempt on this problem — they show that if gradient magnitudes projected onto subspaces decay rapidly, the empirical loss of NoisySGD becomes independent to the model dimension. However, the exact behaviors of gradients remain intractable to analyze theoretically, and it remains uncertain whether the "dimension independence" property is robust across different fine-tuning applications.

In this work, we explore NoisySGD behaviors from an alternative direction — we employ an idealized representation of pre-trained models using the neural collapse theory (Papyan et al., 2020a; Fang et al., 2021) and explore the dimension dependence of NoisySGD in a specific private fine-tuning setup — fine-tuning only the last layer of the pre-trained model. To elaborate, neural collapse is a phenomenon observed during representation learning with expressive feature maps, such as those

employed in the pre-training process. From a high-level perspective, this phenomenon suggests that the features from the last year are sufficiently distinct to separate two different classes. For binary classification problems, this means that the feature for the class labeled $y = 1$ is $m_1$ and the feature for the other class labeled with $y = -1$ is $-m_1$. Mathematically, it suggests that the last layer features of individual classes will converge to a $K$-simplex equiangular tight frame (ETF) where $K$ is the number of classes (see Figure 1). Therefore, fine-tuning only the last-layer is equivalent to training a linear model on top of the $K$-ETF features.

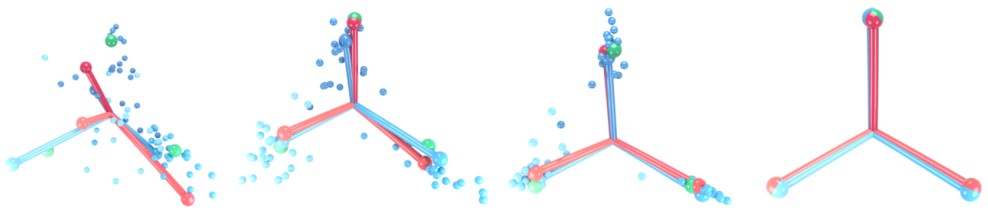

Figure 1: The figure depicts the evolution of the feature layer outputs of a VGG13 neural network when trained on the CIFAR10 with three randomly selected classes. Each class is represented by a distinct color in the small blue sphere. As the training evolves, the last-layer feature mean collapse onto their class. Credit to Papyan et al. (2020b)

Table 1: A summary of NoisySGD behaviors under different private fine-tuning regimes

| Behaviors | Perfect collapse | Fixed perturbations | data normalization |
|---|---|---|---|
| Dimension-independent | ✓ | ✗ | ✓ |
| Robust to Class-imbalance | ✓ | ✗ | ✗ |

Private fine-tuning is vulnerable to fixed perturbations, but our proposed data normalization can effectively mitigate these perturbations.

We investigates the dimension dependence of private fine-tuning both theoretically and empirically. Our contributions are four-folds.

- Theoretically, we show that when feature embeddings exhibit perfect neural collapse, the accuracy of NoisyGD's output is dimension-independent.

- Outside of a perfect neural collapse, if features experience minor perturbations (either fixed or stochastic), the accuracy becomes dimension-dependent again. This highlights that DP fine-tuning lacks robustness against perturbations.

- Empirically, we show that fine-tuning an ImageNet pre-trained vision transformer is not affected by the last-layer dimension on CIFAR-10. However, we observe a degradation in the utility-dimension trade-off when perturbations are introduced, aligning with our theoretical results.

- We adopt data normalization and PCA to address the non-robustness issues of DP fine-tuning. Specifically, for fixed perturbations, our findings suggest that applying data normalization to private data effectively makes it robust to fixed perturbation, eliminating the dimension-dependence issue. In the case of random perturbations, we theoretically demonstrate that PCA is effective to mitigate the dimension dependency.

## 2 PRELIMINARIES

### 2.1 DIFFERENTIAL PRIVACY

In the context of machine learning, DP requires the output of the learning algorithm to be indistinguishable upon addition or removal of an individual training data point. Mathematically, let $\mathcal{D} = \{z_i\}_{i=1}^n \subset \mathcal{Z}$ be a fixed dataset of size $n$ and consider a randomized algorithm $\mathcal{A} : \mathcal{Z}^n \to \mathcal{S}$

| Feature | Sample Complexity |
|---|---|
| Perfect Neural Collapse | $\frac{2\sqrt{\log(1/\gamma)}}{\sqrt{\rho}}$ |
| $\beta$-stochastic perturbation (test) | $\frac{\max\{\sqrt{p}\beta,1\}\sqrt{\log(1/\gamma)}}{\sqrt{2\rho}}$ |
| $\beta$-adversarial perturbation (test) | $\frac{\max\{p\beta,1\}\sqrt{\log(1/\gamma)}}{\sqrt{2\rho}}$ |
| $\beta$-offset perturbation (training) | $\frac{\max\{\sqrt{p}\beta,1\}\sqrt{\log(1/\gamma)}}{\sqrt{2\rho}}$ |
| $\beta$-offset NC + $\alpha$-Class imbalance | $\frac{\max\{\sqrt{p}\beta,1\}\sqrt{\log(1/\gamma)}}{(1-\beta+2\beta\alpha)\sqrt{2\rho}}$ |

Table 2: Summary of the sample complexity upper bounds of achieving a $\gamma$-accuracy of private learning under $\rho$-zCDP. We consider both the perfect and perturbed neural collapse setting. $p$ represents the model dimension.

that maps a dataset $\mathcal{D}$ to $\mathcal{A}(\mathcal{D})$ in some probability space $\mathcal{S}$. We say $\mathcal{A}$ satisfies $(\epsilon, \delta)$-DP for some $\epsilon \geq 0$ and $0 \leq \delta \leq 1$ if

$$\mathbb{P}[\mathcal{A}(\mathcal{D}_0) \in S] \leq e^\epsilon \mathbb{P}[\mathcal{A}(\mathcal{D}_1) \in S] + \delta,$$

for any event $S \in \mathcal{S}$ and any neighboring datasets $\mathcal{D}_0$ and $\mathcal{D}_1$.

Our analysis is based on $\rho$-zCDP which is defined based on the Rényi divergence between $\mathcal{A}(\mathcal{D}_0)$ and $\mathcal{A}(\mathcal{D}_1)$. The Rényi divergence of order $\widetilde{\alpha} > 1$ between $P$ and $Q$ is given by

$$R_{\widetilde{\alpha}}(P\|Q) = \frac{1}{\widetilde{\alpha} - 1} \log \int \left(\frac{p(x)}{q(x)}\right)^{\widetilde{\alpha}} q(x)dx.$$

For $\widetilde{\alpha} = 1$ or $+\infty$, $R_1$ or $R_\infty$ is the limit of $R_{\widetilde{\alpha}}$ as $\widetilde{\alpha}$ tends to 1 or $+\infty$. With a little bit abuse of notations, we define the Rényi divergence between two random variables as the divergence between their distributions. Then, an algorithm is said to satisfy $\rho$-zCDP if $R_{\widetilde{\alpha}}(\mathcal{A}(\mathcal{D}_0)\|\mathcal{A}(\mathcal{D}_1)) \leq \rho\widetilde{\alpha}$ for any neighboring datasets $\mathcal{D}_0$ and $\mathcal{D}_1$ and any $1 < \widetilde{\alpha} < \infty$.

## 2.2 NEURAL COLLAPSE AND PRIVATE FINE-TUNING

**Private fine-tuning setup.** We focus on fine-tuning the last-layer of pre-trained models using the NoisyGD algorithm, which has consistently achieved state-of-the-art results across both vision and language classification tasks (Tramer & Boneh, 2020; De et al., 2022). Consider a $K$-class classification problem with $\mathcal{Z}$ being a probability space. Each data point $z \in \mathcal{Z}$ can be rewritten as $z = (x, y)$ with $x \in \mathbb{R}^p$ being the feature and $y = (y_1, \cdots, y_K) \in \{0,1\}^K$ being generated by the one-hot encoding, that is $y$ belongs to the $k$-th class if $y_k = 1$ and $y_j = 0$ for $j \neq k$.

Under the regime of finetuning, only the last-layer parameter is trained by NoisyGD, which is linear in terms of the features. Thus, we consider the model $f_W(x) = Wx$ with $W \in \mathbb{R}^{K \times p}$ being the last-layer parameter to be trained. Let $\ell : \mathbb{R}^K \times \mathbb{R}^K \to \mathbb{R}$ be a loss function that maps $f_W(x) \in \mathbb{R}^K$ and the label $y$ to $\ell(f_W(x), y)$. For example, for the cross-entropy loss, we have $\ell(f_W(x), y) = -\sum_{i=1}^K y_i \log[(f_W(x))_i]$.

**Neural collapse.** Neural collapse demonstrates that the input feature of the last layer converges to the column of an equiangular tight frame (ETF). Mathematically, an ETF is a matrix

$$M = \sqrt{\frac{K}{K-1}} P \left(I_K - \frac{1}{K}\mathbf{1}_K\mathbf{1}_K^T\right) \in \mathbb{R}^{p \times K},$$

where $P = [P_1, \cdots, P_K] \in \mathbb{R}^{p \times K}$ is a partial orthogonal matrix such that $P^T P = I_K$. Here for a given dimension $d = p$ or $K$, we denote $I_d \in \mathbb{R}^d$ the identity matrix and denote $\mathbf{1}_d = [1, \cdots, 1]^T \in \mathbb{R}^d$. Denote $M = [M_1, \cdots, M_K]$ with $M_k$ being the $k$-th column of $M$. We define the classification problem under perfect neural collapse as follows.

**Definition 1** (Classification problem under prefect neural collapse)**.** *Let there be $K$ classes. The distribution $\mathbb{P}[x = M_k|y_k = 1] = 1$ for $k = 1, ..., K$.*

Denote $\mathcal{N}(0, \sigma^2 I_d)$ a Gaussian random variable in $\mathbb{R}^d$ with covariance matrix $\sigma^2 I_d$. Let $\theta \in \mathbb{R}^p$ be the vectorization of the parameters. With a little bit abuse of notations, we rewrite $\ell_i(\theta) = \ell(\theta, z_i)$ the loss function with respect to the data point $z_i$. The noisy gradient descent is defined as follows.

**Noisy Stochasitic Gradient Descent (NoisySGD).** Let the loss function $\mathcal{L}(\theta) := \sum_{i=1}^n \ell(\theta, z_i)$. The noisy gradient descent algorithm outputs

$$\theta_{t+1} = \theta_t - \frac{\eta_t}{|\mathcal{B}_t|} \left( \sum_{z_i \in \mathcal{B}_t} \max\left\{1, \frac{G}{\|\nabla\ell_i(\theta_t)\|_2}\right\} \nabla\ell_i(\theta_t) + \mathcal{N}\left(0, \frac{G^2}{2\rho} I_d\right) \right).$$

Here $\mathcal{B}_t$ is a mini-batch selected uniformly at random in the $t$-th step and the gradient is clipped by a constant $G > 0$ to control the sensitivity of the gradient. The algorithm that runs the above for $T$ iterations satisfies $T\rho$-zCDP (Abadi et al., 2016).

When $\mathcal{B}_t$ represents the entire training set, the NoisySGD algorithm reduces to a Noisy Gradient Descent (NoisyGD) algorithm.

**Noisy Gradient Descent (NoisyGD).** Let the loss function $\mathcal{L}(\theta) := \sum_{i=1}^n \ell(\theta, z_i)$. The noisy gradient descent algorithm outputs

$$\theta_{t+1} = \theta_t - \eta_t \left( \sum_{i=1}^n \max\left\{1, \frac{G}{\|\nabla\ell_i(\theta_t)\|_2}\right\} \nabla\ell_i(\theta_t) + \mathcal{N}\left(0, \frac{G^2}{2\rho} I_d\right) \right). \tag{1}$$

Here the gradient is clipped by a constant $G > 0$ to control the sensitivity of the gradient. The algorithm that runs the above for $T$ iterations satisfies $T\rho$-zCDP (Abadi et al., 2016).

## 3 DP-LEARNING UNDER NEURAL COLLAPSE

In this section, we present the theoretical results for DP-learning under neural collapse. In the case of perfect neural collapse, we demonstrate that the accuracy is independent of the dimension $p$. Subsequently, we delve into the misclassification error analysis for the more realistic imperfect neural collapse, where the features are perturbed.

### 3.1 PERFECT NEURAL COLLAPSE

Recall that under perfect neural collapse, we assume $\mathbb{P}[x = M_k|y_k = 1] = 1$. Let $\widehat{\theta} \in \mathbb{R}^{Kp}$ be the 1-step output of the noisy GD algorithm equation 1 with 0-initialization and let $\widehat{W} \in \mathbb{R}^{K \times p}$ be the matrix derived from $\widehat{\theta}$. Denote $\widehat{y} = \text{OneHot}(\widehat{W}x) \in \{0, 1\}^K$ the predictor after the one-hot encoding, that is $\widehat{y}_i = 1$ if $i = \arg\max_j\{(\widehat{W}x)_j\}$, otherwise $\widehat{y}_i = 0$.

**Theorem 2.** *Let $\widehat{y}$ be a predictor trained by NoisyGD under the cross entropy loss with zero initialization. Assume that the training dataset is balanced, that is, the sample size of each class is $n/K$. For classification problems under neural collapse with $K$ classes, the mis-classification error is*

$$\mathbb{P}[\widehat{y} \neq y] = (K-1)\Phi\left(-\frac{n}{K\sigma}\left(1 + \frac{K-2}{K(K-1)}\right)\right)$$

$$\leq (K-1)e^{-\frac{C_K n^2}{K\sigma^2}}$$

*with $\sigma^2 = \frac{G^2}{2\rho}$ and $C_K = \left(1 + \frac{K-2}{K(K-1)}\right)^2$. As a result, to achieve the $\gamma$-accuracy, the sample complexity is $O\left(\frac{\sqrt{\log(1/\gamma)}}{\sqrt{\rho}}\right)$.*

The theorem offers several insights, we have

1. The error bound is exponentially close to 0 if $\rho \gg G^2/n^2$ — very strong privacy and very strong utility at the same time.
2. The result is dimension independent — it doesn't depend on the dimension $p$.

3. The result is robust to class imbalance for $K = 2$, if we apply a re-parameterization of private data (see more details below).

4. The result is independent of the shape of the loss functions. Logistic loss works, while square losses also works.

5. The result does not require careful choice of learning rate. Any learning rate works equally well.

One may refer to Appendix A for the proof of Theorem 2 and corresponding insights. In Theorem 2, we have a class-balance assumption. However, when $K = 2$ — where the cross entropy loss aligns with the logistic loss — this assumption can be removed by re-parameterization. Precisely, an equivalent neural collapse case gives $M = [-e_1, e_1]$ with $e_1 = [1, 0, \ldots, 0]^T$. Furthermore, we consider the re-parameterization with $y \in \{-1, 1\}$, $\theta \in \mathbb{R}^p$ and the decision rule being $\hat{y} = \text{sign}(\theta^T x)$. Then, the logistic loss is $\log(1 + e^{-y \cdot \theta^T x})$.

**Theorem 3.** *For the case $K = 2$, under perfect neural collapse, using the aforementioned re-parameterization, the sample complexity of one step NoisyGD to achieve $1 - \delta$ accuracy under $\rho$-zCDP is $O\left(\frac{\sqrt{\log(1/\delta)}}{\sqrt{\rho}}\right)$.*

Our dimension-independent findings can be extended to the domain adaptation context.

**Neural Collapse in Domain Adaptation:** In many private fine-tuning scenarios, the model is initially pre-trained on an extensive dataset with thousands of classes (e.g., ImageNet), denoted as $K_0$ class, and is subsequently fine-tuned for a downstream task with a smaller number of classes, denotes as $K \leq K_0$. We formalize it under the neural collapse setting as follows. Let $\widetilde{M} = [\widetilde{M}_1, \cdots, \widetilde{M}_K]$ be a matrix where each $\widetilde{M}_i$ is a column of an ETF $M \in \mathbb{R}^{p \times K_0}$. With prefect neural collapse, we assume $\mathbb{P}[x = \widetilde{M}_k | y_k = 1] = 1$. The following theorem shows that the dimension-independent property still holds when private dataset has a subset classes of the pre-training dataset.

**Theorem 4.** *Let $\hat{y}$ be a predictor trained by NoisyGD under the cross entropy loss with zero initialization. Assume that the training dataset is balanced. For multi-class classification problems under neural collapse with $K$ classes, subset of a gigantic dataset with $K_0 \geq K$ classes, the sample complexity to achieve $(1 - \gamma)$-accuracy is $O\left(\frac{\sqrt{\log(1/\gamma)}}{\sqrt{\rho}}\right)$.*

The proof of Theorem 4 is given in Appendix A.3.

### 3.2 NON-ROBUSTNESS TO PERTURBATIONS IN TESTING DATA

The properties of NoisyGD are rather surprising especially given how delicate DP learning is in general without neural collapse. However, let us now show that if we wiggle even a little bit from exact neural collapse, we lose the amazingly strong performance. In this section, we investigate the effect of introducing minor perturbations to testing data and observe that NoisyGD is no longer dimension-free. For simplicity, we investigate the case with $K = 2$ and consider the re-parameterization introduced in Section 3.1.

**Non-robustness to perturbation in test time.** Assume that, in prediction time, the feature embeddings are perturbed by a small value in $\ell_\infty$, i.e., each feature embedding has the form $x_i = y_i e_1 + v$, where $v = [v_1, v_2, \ldots, v_p]^T$ with $\|v\|_\infty \leq \beta$ is a fixed perturbation. Then the following bound reveals that the dimension dependence shows up and the results becomes very fragile.

1. Fixed perturbation: If we allow the perturbation $v$ to be adversarially chosen, then there exists $v$ satisfying $\|v\|_\infty \leq \beta$ such that the sample complexity bound to achieve $(1 - \gamma)$-accuracy is $O\left(\frac{G \max\{p\beta, 1\}\sqrt{\log(1/\gamma)}}{\sqrt{2\rho}}\right)$.

2. Stochastic Perturbation: If the perturbation is stochastic and is drawn by randomly sampling from $\{-\beta, \beta\}$ with probability $0.5$ for each, then we still require $O\left(\frac{\max\{\sqrt{p}\beta, 1\}\sqrt{\log(1/\gamma)}}{\sqrt{2\rho}}\right)$ samples to achieve $1 - \gamma$ accuracy.

### 3.3 NON-ROBUSTNESS TO PERTURBATIONS IN TRAINING DATA

In this section, we explore scenarios in which the training feature is not pre-trained to perfection and experiences a fixed offset perturbation. Furthermore, we note that the presence of class imbalance makes NoisyGD's output more susceptible to perturbations. The proofs of sample complexities are given in Appendix C.

**Non-robustness to offset in the class-balanced case.** Even if we just shift the training feature vectors away by a constant offset (while keeping the same margin), it makes DP learning a lot harder. Note that in the class-balanced case, this makes absolutely no difference to the gradient, when we start from 0 because

$$\nabla \mathcal{L}(\theta) = \frac{n}{2} \cdot 0.5 \cdot -(-e_1 + v) + \frac{n}{2} \cdot 0.5 \cdot (e_1 + v) = \frac{n}{2} e_1.$$

However, for the private learning problem, the sensitivity becomes much larger. If we want clipping to remain inactive, $G$ needs to be chosen to be larger than $\sqrt{1 + \|v\|_2^2}$. Or if we still choose $G$ to be 1, then every data point needs to be shrunk to $(e_1 + v)/\sqrt{1 + \|v\|_2^2}$. In either case, if all we know is that $\|v\|_\infty \leq \beta$, the sample complexity for achieving $1 - \gamma$ classification error will be proportional to $\|v\| = O(\sqrt{p}\beta)$, i.e., DP-SGD is no longer dimension-free as demonstrated in the following proposition.

If both the training data and testing data are shifted by an offset term $v$, the dimension dependency is more severe. In fact, the sample complexity to achieve $1 - \gamma$ accuracy is $O\left( \frac{\max\{p\beta^2, 1\}\sqrt{\log(1/\gamma)}}{\sqrt{\rho}} \right)$.

**Non-robustness to class imbalance.** Note that in the above case, it is quite a coincidence that $v$ gets cancelled out in the non-private gradient. Either when the class is not balanced or when the initialization is not 0, the offset $v$ will be part of the gradient that overwhelms the signal. Consider the case where we have $\alpha n$ data points with label $-1$ and $(1 - \alpha)n$ data points with label 1 for $\alpha \neq 0.5$, and we start at 0, then

$$\nabla \mathcal{L}(\theta) = \alpha n \cdot 0.5 \cdot -(-e_1 + v) + (1 - \alpha)n \cdot 0.5 \cdot (e_1 + v) = \frac{n}{2} e_1 + \frac{(1 - 2\alpha)n}{2} v.$$

Suppose that we have $\alpha n$ data points with label $-1$ and $(1-\alpha)n$ data points with label 1 for $\alpha \neq 0.5$. If we allow the perturbation $v$ to be adversarially chosen, then there exists $v$ satisfying $\|v\|_\infty \leq \beta$ such that the sample complexity bound to achieve $1 - \gamma$ robust classification under neural collapse is $O\left( \frac{\max\{\sqrt{p}\beta, 1\}\sqrt{\log \frac{1}{\delta}}}{\sqrt{(1 - \beta + 2\beta\alpha)^2 \cdot \rho}} \right)$.

**Random perturbation.** Denote $\{v_i\}_{i=1}^n \subseteq \mathbb{R}^p$ a sequence of i.i.d. copies of a random vector $v$. Each component of $v$ takes on the value of $\pm\beta$ with a probability 0.5. Consider the binary classification problem with training set $\{(x_i, y_i)\}_{i=1}^n$. Here $x_i = e_1 + v_i$ if $y_i = 1$ and $x_i = -e_1 + v_i$ if $y_i = -1$. Then, the sample complexity to achieve $(1 - \gamma)$-accuracy is $O\left( \sqrt{\frac{p\beta^2 \log(1/\gamma)}{\rho}} \right)$.

## 4 SOLUTIONS FOR NON-ROBUSTNESS ISSUES

In this section, we explore various solutions to address the dimension dependency arising from perturbed features. For fixed perturbations, we consider releasing the mean of feature embeddings to cancel out the perturbation. In the case of random perturbations, we suggest to carry out dimension reduction.

### 4.1 ADDRESSING FIXED PERTURBATION: NORMALIZATION AND DIFFERENCING

Consider the training set $\{(x_i, y_i)\}_{i=1}^n$ and denote $X_k = \{x_i : (x_i, y_i)$ is in the $k$-th class$\}$. Recall the case where the feature is shifted by a constant offset $v$, where one has $x_i = \widetilde{M}_k = M_k + v$ for $x_i \in X_k$.

To deal with the offset $v$, we pre-process the feature as $\widetilde{x}_i = x_i - \frac{1}{n} \sum_{j=1}^{n} x_j$. Then, if the class is balanced, it holds $\widetilde{x}_i = \widetilde{M}_k - \frac{1}{K} \sum_{j=1}^{K} \widetilde{M}_j = M_k$ for $x_i \in X_k$. That is, the perturbations canceled out.

We still need to bound the sensitivity of the gradient when training with $\{\widetilde{x}_i, y_i\}_{i=1}^{n}$. If we delete arbitrary $(x_j, y_j)$ from the dataset, for the case $K = 2$ with data balance, then the sensitivity $G$ of the gradient can be bounded as

$$G = \left\| \sum_{i=1}^{n} y_i \left( x_i - \frac{1}{n} \sum_{j=1}^{n} x_j \right) - \sum_{l \neq j} y_l \left( x_l - \frac{1}{n-1} \sum_{m \neq j} x_m \right) \right\|$$

$$= \left\| x_j y_j - \frac{y_j}{n-1} \sum_{m \neq j} x_m \right\|_2 = \frac{n}{n-1},$$

where the second equality is because $\sum_{i=1}^{n} y_i = 0$ thanks to the data balance. Since $G \leq 2$ is upper bounded by a dimension-independent constant, the sample complexity to achieve $(1-\gamma)$-accuracy is $O\left( \frac{\sqrt{\log(1/\gamma)}}{\sqrt{\rho}} \right)$.

Note that this normalization method is not robust to class imbalance. In fact, if we consider the class imbalanced case with which we have $\alpha n$ data points with label $+1$ and the rest $(1-\alpha)n$ data points with label $-1$ for some $\alpha > 0$, then we have $\widetilde{x}_i = 2(1-\alpha)e_1$ for $y_i = 1$ and $\widetilde{x}_i = -2\alpha e_1$ for $y_i = -1$. In this class-imbalance case, one can recover the feature embedding $e_1$ and $-e_1$ by considering $\frac{\widetilde{x}_i}{\|\widetilde{x}_i\|_2}$. Howoever, in this case, the sensitivity remains a constant $G_\alpha$ which, although independent of the dimension $p$, still relies on $\alpha$.

## 4.2 MITIGATING RANDOM PERTURBATION: DIMENSION REDUCTION

In Abadi et al. (2016), dimension reduction methods such as DP-PCA has been adopted to enhance the model performance. In this section, we investigate the theory to explain the success of dimension reduction methods under the regime of neural collapse.

Denote $\{v_i\}_{i=1}^{n} \subseteq \mathbb{R}^p$ a sequence of i.i.d. copies of a random vector $v$. Recall the binary classification problem $(K = 2)$ with training set $\{(x_i, y_i)\}_{i=1}^{n}$. Here $x_i = e_1 + v_i$ if $y_i = 1$ and $x_i = -e_1 + v_i$ if $y_i = -1$. As discussed in Section 3.3, even in the class balanced case, the accuracy is dimension-dependent as the sensitivity of the gradient depends on $p$.

To deal with the dimension-dependency, we consider the dimension reduction methods. Precisely, we aim to generate a projection matrix $\widehat{P} = [\widehat{P}_1, \cdots, \widehat{P}_{K-1}] \in \mathbb{R}^{p \times (K-1)}$ and training with $\{(\widetilde{x}_i = \widehat{P}x_i, y_i)\}_{i=1}^{n}$. It is easy to see that the "best projection" is such that $\text{span}(\widehat{P})$ is the same as the space spanned by $\{M_i\}_{i=1}^{K}$.

In practice, one can not obtain $\{M_i\}_{i=1}^{K}$ and $\widehat{P}$ can be generated by some public dataset $\{(\widehat{x}_i, \widehat{y}_i)\}_{i=1}^{m}$, such as the public dataset used in the pre-training process. In Section 4.3, we will discuss generating the projection matrix by averaging the feature or using PCA. In some scenarios without a public dataset, one can generate $\widehat{P}$ using the training dataset with some DP algorithms, such as DP-PCA (Abadi et al., 2016; Liu et al., 2022).

For $K = 2$, generating the projection matrix means we aim to generate a vector $\widehat{P}$ to approximate $e_1$. Consider a projection vector $\widehat{P} = e_1 + \Delta$ with some perturbation $\Delta$ satisfying $\|\Delta\|_\infty \leq \beta_0$. The following theorem says that for $\beta_0 < \frac{1}{p}$, the mis-classification error is dimension independent.

**Theorem 5.** *For the NoisyGD trained with $\{\widetilde{x}_i, y_i\}$, the sample complexity to achieve $(1-\gamma)$-accuracy is*

$$n = O\left( \sqrt{\frac{G_{\beta, \beta_0, p}^2 \log \frac{2}{\gamma}}{M_{\beta, \beta_0, p} \rho}} \right)$$

with $G_{\beta,\beta_0,p} = 1 + \beta(1 + \beta_0 + p\beta_0)$ and $M_{\beta,\beta_0,p} = (1 - \beta_0)^2 - p\beta\beta_0 - (1 + \beta_0)(\beta + \beta_0 p) - (\beta + \beta_0 p)(1 + \beta + \beta_0 + \beta\beta_0 p)$. Moreover, if we assume that $\beta_0 \le 1/p$, then the sample complexity is dimension independent, that is, the effect of dimension $p$ is canceled out by $\beta_0$.

### 4.3 CONSTRUCTION OF THE PROJECTION MATRIX

The next question is to construct the projection matrix $\widehat{P} = [\widehat{P}_1, \cdots, \widehat{P}_{K-1}] \in \mathbb{R}^{p \times (K-1)}$. Let $\{(\widehat{x}_i, \widehat{y}_i)\}_{i=1}^m$ be a set of public dataset. Assume that $\widehat{x}_i = M_k + \widehat{v}_i$ for $\widehat{y}_i$ in the $k$-th class. Here $\widehat{v}_i$ are i.i.d. copies of a bounded zero-mean random variable $v \in \mathbb{R}^p$ with covariance matrix $\mathbb{E}[vv^T] = \sigma_0^2 I$. We consider the following to methods two generate the projection matrix.

**Averaging the features.** Denote $X_k = \{\widehat{x}_i : \widehat{y}_i$ belongs to the $k$-th class$\}$. Let $\widehat{P}_j = \frac{1}{m_k}\sum_{\widehat{x}_i \in X_k} \widehat{x}_i$ with $m_k$ being the cardinality of $X_k$ for $1 \le j \le K - 1$. Then, we have $\Delta = \widehat{P}_j - M_j = \frac{1}{m_k}\sum_{\widehat{x}_i \in X_k} \widehat{x}_i$. By the concentration inequality, we have $\beta_0 = \|\Delta\|_\infty \le O\left(\frac{\sigma_0}{\sqrt{m_k}}\right)$ with probability $pe^{-m_k^2}$.

**Principle component analysis.** Let $\{\widehat{P}_j\}_{j=1}^{K-1}$ be the the eigenvectors corresponding to $K - 1$ largest eigenvalues of $\widehat{\Sigma} = \frac{1}{m}\sum_{i=1}^m \widehat{x}_i \widehat{x}_i^T$. Note that under neural collapse $\widehat{\Sigma}$ converges to $\Sigma = e_1 e_1^T + \epsilon^2 I_p$ whose eigenvector corresponding to the largest eigen-value is the feature mean $e_1$. Then, thanks to the theory of matrix perturbation such as the Davis-Kahan Theorem , the space spanned by $\{\widehat{P}_j\}_{j=1}^{K-1}$ can be close to the space spanned by $\{M_i\}_{i=1}^K$ if the eigen-gap is large enough. As $\beta_0$ is the infinity norm of the perturbations, we use a bound on the infinity norm of eigenvectors (Fan et al., 2017). We state the results for $K = 2$ and the proof is similar to (Fan et al., 2017) as the eigengap here is 1. Precisely, let $\widehat{P}$ be the eigenvector of $\frac{1}{m}\sum_{i=1}^m \widehat{x}_i \widehat{x}_i^T$ that corresponds to the largest eigenvalue. Then, it holds

$$\beta_0 = \|\widehat{P} - e_1\|_\infty \le O\left(\frac{1}{\sqrt{m}}\right)$$

with probability $O\left(pe^{-m^2}\right)$.

## 5 EXPERIMENTS

In this section, we conduct experiments to empirically investigate the behavior of NoisyGD (with fine-tuning the last layer) under different robustness settings.

### 5.1 FINE-TUNE NOISYGD WITH SYNTHETIC NEURAL COLLAPSE FEATURE

We first generate a synthetic data matrix $X \in \mathcal{R}^{n \times d}$ with feature dimension $d$ under perfect neural collapse. The number of classes $K$ is 10 and the sample size is $n = 10^4$. In the default setting, we assume each class draws $n/K$ data from a column of $K$-ETF, the training starts from a zero weight $\theta$ and the testing data are drawn from the same distribution as $X$. The Gaussian noise is selected such that the NoisyGD is $(1, 10^{-4})$-DP.

In Figure 2(a), we observe that an imbalanced class alone does not affect the utility. However, NoisyGD becomes non-robust to class imbalance when combined with a private feature offset with $\|\nu\|_\infty = 0.1$. Additionally, it is non-robust to perturbed test data with $\|\nu\|_\infty = 0.1$.

### 5.2 FINE-TUNE NOISYSGD WITH REAL DATASETS

In this section, we empirically investigate the non-robustness of neural collapse using real datatsets. Precisely, we fine-tune NoisySGD with the ImageNet pre-trained vision transformer (Dosovitskiy et al., 2020) on CIFAR-10 for 10 epochs. Test features in the perturb setting are subjected to Gaussian noise with a variance of 0.1. The vision transformer produces a 768-dimensional feature for each image. To simulate different feature dimensions, we randomly sample a subset of coordinates or make copies of the entire feature space.

In Figure 2(b), we observe that while perturbing the testing features degrades the utility of both Linear SGD and NoisySGD, Linear SGD is generally unaffected by the increasing dimension. On the other hand, the accuracy of NoisySGD deteriorates significantly as the dimension increases.

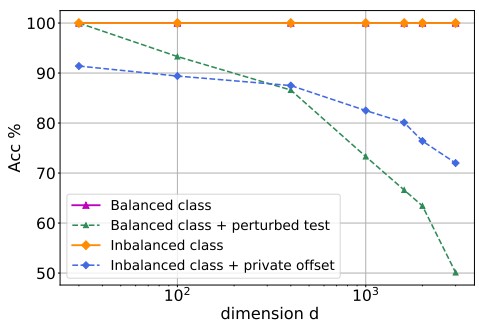
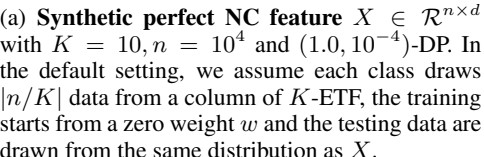
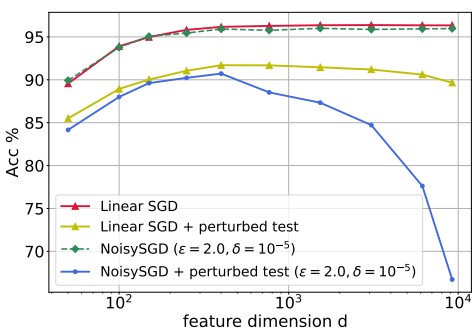

(a) **Synthetic perfect NC feature** $X \in \mathcal{R}^{n \times d}$ with $K = 10, n = 10^4$ and $(1.0, 10^{-4})$-DP. In the default setting, we assume each class draws $\lfloor n/K \rfloor$ data from a column of $K$-ETF, the training starts from a zero weight $w$ and the testing data are drawn from the same distribution as $X$.

(b) **CIFAR-10**: Test features in the perturb setting are subjected to Gaussian noise with a variance of 0.1. The vision transformer produces a 768-dimensional feature for each image. To simulate different feature dimensions, we randomly sample a subset of coordinates or make copies of the entire feature space.

Figure 2: Empirical behaviors of NoisyGD under various robustness setting.

## 6 DISCUSSIONS AND FUTURE WORK

Most existing theory of DP-learning focuses on suboptimality in surrogate loss of testing data. Our paper studies 0-1 loss directly and observed very different behaviors under perfect and near-perfect neural collapse. In particular, we have $\log(1/\text{error})$ sample complexity rather than $1/\text{error}$ sample complexity. Though neural collapse is a strong assumption, it suggests that privacy theorists should look into structures in data and how one can adapt to them. Additionally, our result suggests a number of practical mitigations to make DP-learning more robust in nearly neural collapse settings. It will be useful to investigate whether the same tricks are useful for private learning in general even without neural collapse. Moreover, our results suggest that under neural collapse, choice of loss functions (square loss vs CE loss) do not matter very much for private learning. Square loss has the advantage of having a fixed Hessian independent to the parameter, thus making it easier to adapt to strong convexity parameters like in AdaSSP (Wang, 2018). This is worth exploring.

Noisy-GD and NoisySGD theory suggests that one needs $\Omega(n^2)$ time complexity to achieve optimal privacy-utility-trade off in DP-ERM (faster algorithms exist but more complex and they handle only some of the settings). Our results on the other hand, suggest that when there are structures in the data, e.g., near-perfect neural collapse, the choice of number of iterations is no longer important, thus making computation easier.

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

# A Proofs of Section 3.1

## A.1 Proof of Theorem 2 and corresponding results

Recall an ETF defined by

$$M = \sqrt{\frac{K}{K-1}} P \left( I_K - \frac{1}{K} \mathbf{1}_K \mathbf{1}_K^T \right) = \sqrt{\frac{K}{K-1}} \left( P - \frac{1}{K} \sum_{k=1}^{K} P_k \mathbf{1}_K^T \right),$$

where $P = [P_1, \cdots, P_K] \in \mathbb{R}^{p \times K}$ is a partial orthogonal matrix with $P^T P = I_K$. Rewrite $M = [M_1, \cdots, M_K]$. Let the label $y = (y_1, \cdots, y_K)^T \in \{0, 1\}^K$ be represented by the one-hot encoding, that is, $y_k = 1$ and $y_j = 0$ for $j \neq k$ if $y$ belongs to the $k$-th class.

**Definition 6** (Classification problem under Neural Collapse). *Let there be $K$ classes. The distribution $\mathbb{P}[x = M_k | y_k = 1] = 1$ for $k = 1, ..., K$.*

*Proof of Theorem 2.* Let $W = [W_1, \cdots, W_K]^T \in \mathbb{R}^{K \times p}$. Consider the output function $f_W(x) = Wx \in \mathbb{R}^K$. Suppost that $y_k = 1$. Then, the cross-entropy loss is defined by

$$\ell(f_W(x), y) = -\log \left( \frac{e^{W_k^T x}}{\sum_{k'=1}^{K} e^{W_{k'}^T x}} \right).$$

The corresponding empirical risk is

$$R_n(M, W) = \sum_{k=1}^{K} -n_k \log \left( \frac{e^{W_k^T M_k}}{\sum_{k'=1}^{K} e^{W_{k'}^T M_k}} \right).$$

Note that

$$\nabla_W \ell(f_W(x), y) = (\text{SoftMax}(f_W(x)) - y) \, x^T,$$

where $\text{SoftMax} : \mathbb{R}^K \to \mathbb{R}^K$ is the SoftMax function defined by

$$\text{SoftMax}(z)_i = \frac{e^{z_i}}{\sum_{j=1}^{K} e^{z_j}}, \qquad \text{for all } z \in \mathbb{R}^K.$$

We obtain

$$\nabla_W R_n(M, W) = \sum_{k=1}^{K} n_k \left( \text{SoftMax}(f_W(M_k)) - y^k \right) M_k^T,$$

where $y^k$ is the label of the $k$-th class. For zero initialization, we have

$$\text{SoftMax}(f_{\mathbf{0}}(M_k)) = \frac{1}{K} \mathbf{1}_K$$

and

$$\nabla_W (R_n(M, W)) \Big|_{W=\mathbf{0}} = \sum_{k=1}^{K} n_k \left( \frac{1}{K} \mathbf{1}_K - y^k \right) M_k^T. \tag{2}$$

Now we consider one step NoisyGD from 0 with learning rate $\eta = 1$:

$$\widehat{W} = -\sum_{k=1}^{K} n_k \left( \frac{1}{K} \mathbf{1}_K - y^k \right) M_k^T + \Xi,$$

where $\Xi \in \mathbb{R}^{K \times p}$ with $\Xi_{ij}$ drawn independently from a normal distribution $\mathcal{N}(0, \sigma^2)$.

Consider $x = M_k$. It holds

$$f_{\widehat{W}}(x) = \widehat{W} M_k = -\sum_{k'=1}^{K} n_{k'} \left( \frac{1}{K} \mathbf{1}_K - y^{k'} \right) M_{k'}^T M_k + \Xi M_k.$$

Since

$$\Xi M_k \sim \mathcal{N}\left(0, \sigma^2 \|M_k\|_2^2 I_K\right) \qquad \text{and} \qquad \|M_k\|_2^2 = 1$$

we have

$$\widehat{W} M_k \sim \mathcal{N}\left(\boldsymbol{\mu}_{n,K}, \sigma^2 I_K\right),$$

where $\boldsymbol{\mu}_{n,K} = -\sum_{k'=1}^{K} n_{k'} \left(\frac{1}{K}\mathbf{1}_K - y^{k'}\right) M_{k'}^T M_k$. Note that

$$M_{k'}^T M_k = \frac{K}{K-1}\left(\delta_{k,k'} - \frac{1}{K}\right).$$

We obtain

$$(\boldsymbol{\mu}_{n,K})_j = \begin{cases} n/K, & j = k, \\ -\frac{n(K-2)}{K^2(K-1)}, & j \neq k, \end{cases}$$

for $n_{k'} = n/K$ (balanced data). By the union bound, the mis-classification error is

$$(K-1)\mathbb{P}\left[\mathcal{N}(n/K, \sigma^2) < \mathcal{N}(-\frac{n(K-2)}{K^2(K-1)}, \sigma^2)\right] = (K-1)\Phi\left(-\frac{n}{K\sigma}\left(1 + \frac{K-2}{K(K-1)}\right)\right)$$

$$\square$$

**Proof sketches of the insights.** Note that in Equation equation 2, the gradient is a linear function of the feature map thanks to the zero-initialization while for least-squares loss, one can derive a similar gradient as Equation equation 2. Thus, the proof can be extended to the least squares loss directly. Moreover, by replacing $n_k$ with $n_k\eta$ in equation 2, one can extend the results to any $\eta$.

### A.2 PROOF OF THEOREM 3

Recall the re-parameterization for $K = 2$. Precisely, an equivalent neural collapse case gives $M = [-e_1, e_1]$ with $e_1 = [1, 0, \ldots, 0]^T$. Furthermore, we consider the re-parameterization with $y \in \{-1, 1\}$, $\theta \in \mathbb{R}^p$ and the decision rule being $\hat{y} = \text{sign}(\theta^T x)$. Then, the logistic loss is $\log(1 + e^{-y \cdot \theta^T x})$.

*Proof of Theorem 3.* According to the re-parameterization, for the class imbalanced case, we have

$$\hat{\theta} = -\eta\left(\frac{n}{2} \cdot 0.5 \cdot (-\begin{bmatrix} -1 \\ 0 \\ \vdots \\ 0 \end{bmatrix}) + \frac{n}{2} \cdot 0.5 \cdot \begin{bmatrix} 1 \\ 0 \\ \vdots \\ 0 \end{bmatrix} + \mathcal{N}(0, \frac{G^2}{2\rho}I_p)\right) = -\eta\left(\begin{bmatrix} n/2 \\ 0 \\ \vdots \\ 0 \end{bmatrix} + \mathcal{N}(0, \frac{G^2}{2\rho}I_p)\right).$$

The rest of the proof is similar to that of Theorem 2.

For the class-imbalanced case, assume that we have $\alpha n$ data points have with label $+1$ while the rest $(1-\alpha)n$ points have label $-1$. Then, the gradient is

$$\hat{\theta} = -\eta\left(\frac{n\alpha}{2} \cdot \cdot (-\begin{bmatrix} -1 \\ 0 \\ \vdots \\ 0 \end{bmatrix}) + \frac{n(1-\alpha)}{2} \cdot \cdot \begin{bmatrix} 1 \\ 0 \\ \vdots \\ 0 \end{bmatrix} + \mathcal{N}(0, \frac{G^2}{2\rho}I_p)\right) = -\eta\left(\begin{bmatrix} n/2 \\ 0 \\ \vdots \\ 0 \end{bmatrix} + \mathcal{N}(0, \frac{G^2}{2\rho}I_p)\right).$$

Thus, the same conclusion holds. $\square$

### A.3 PROOF OF THEOREM 4

In this section, we consider a broad pre-training on a gigantic dataset with $K_0$ classes. The downstream task is a $K$-class classification problem with $K \le K_0$. Let $P = [P_1, \cdots, P_{K_0}] \in \mathbb{R}^{p \times K_0}$ be a partial orthogonal matrix with $P^T P = I_{K_0}$. Let

$$M_0 = \sqrt{\frac{K_0}{K_0 - 1}} P \left( I_{K_0} - \frac{1}{K_0} \mathbf{1}_{K_0} \mathbf{1}_{K_0}^T \right) = \sqrt{\frac{K_0}{K_0 - 1}} \left( P - \frac{1}{K_0} \sum_{k=1}^{K_0} P_k \mathbf{1}_{K_0}^T \right).$$

Denote $M = [M_1, \cdots, M_K]$ with each $M_k$ being a column of $M_0$. Note that

$$M_{k'}^T M_k = \frac{K_0}{K_0 - 1} \left( \delta_{k,k'} - \frac{1}{K_0} \right).$$

We have

$$\boldsymbol{\mu}_{n,K} := - \sum_{k'=1}^{K} n_{k'} \left( \frac{1}{K} \mathbf{1}_K - y^{k'} \right) M_{k'}^T M_k.$$

For $j \ne k$, we have

$$(\mu_{n,K})_j = -\frac{n}{K} \left[ \frac{1}{K} + \frac{K-1}{K(K_0-1)} - \frac{K-2}{K(K_0-1)} \right] = -\frac{n(K_0-2)}{K^2(K_0-1)}.$$

For $j = k$, it holds

$$(\mu_{n,K})_j = -\frac{n}{K} \left[ \frac{1}{K} - 1 - \frac{K-1}{K(K_0-1)} \right] = \frac{n(K-1)K_0}{K^2(K_0-1)}.$$

By the union bound, the mis-classification error is

$$(K-1)\mathbb{P}\left[ \mathcal{N}((\mu_{n,K})_k, \sigma^2) < \mathcal{N}((\mu_{n,K})_1, \sigma^2) \right] = (K-1)\Phi\left( \frac{nC_{K,K_0}}{\sigma} \right)$$

with $C_{K,K_0} = \frac{1}{K} \left[ \frac{K \cdot K_0 - 2}{K^2(K_0-1)} \right]$.

## B RESULTS FOR PURTURBING THE TESTING DATA

### B.1 FIXED PERTURBATION

Recall that the output of DP-GD has the form $\widehat{\theta} = \mathcal{N}(-\frac{\eta n}{2}, \sigma^2)$. One has

$$\hat{\theta}^T(e + v) = \frac{n}{2} + \mathcal{N}(0, \frac{G^2(p\epsilon^2 + 1)}{2\rho}).$$

The sample complexity can be derived similarly as previous sections, which is dimension dependent.

### B.2 RANDOM PERTURBATION

Let's say in prediction time, the input data point can be perturbed by a small value in $\ell_\infty$. If we allow the perturbation to be adversarial chosen, then there exits $v$ satisfying $\|v\|_\infty \le \beta$ such that

$$\hat{\theta}^T(x + v) = \frac{n}{2} + \frac{G}{\sqrt{2\rho}} Z_1 - \sum_{i=1}^{p} |Z_i| \frac{G\beta}{\sqrt{2\rho}}$$

where $Z_1, ..., Z_n \sim \mathcal{N}(0,1)$ i.i.d. Note that the additional term scales as $O(p\frac{G\beta}{\sqrt{\rho}})$, which can alter the prediction if $p \asymp n$ even if $\rho$ is a constant (weak privacy).

The number of data points needed to achieve $1 - \delta$ robust classification under neural collapse is $O\left( \frac{G \max\{p\epsilon, 1\} \sqrt{\log(1/\delta)}}{\sqrt{2\rho}} \right)$.

## C    RESULTS FOR PERTURBING THE TRAINING DATA

### C.1    FIXED PERTURBATION

Without loss of generality, we assume $0 < \alpha < 1/2$ Consider the class imbalanced case with $n_{-1} = \alpha n$ and $n_{+1} = (1 - \alpha)n$. The gradient for $\theta_0 = 0$ is

$$\nabla \mathcal{L}(\theta_0) = \alpha n \cdot 0.5 \cdot -(-e_1 + v) + (1 - \alpha)n \cdot 0.5 \cdot (e_1 + v) = \frac{n}{2}e_1 + \frac{(1 - 2\alpha)n}{2}v.$$

Thus, the output is

$$\widehat{\theta} = -\eta \left( \frac{n}{2}e_1 + \frac{(1 - 2\alpha)n}{2}v + \mathcal{N}(0, \sigma^2) \right)$$

The sensitivity is $G = \sqrt{1 + \|v\|_2}$ and $\sigma^2$ is taken to be $G^2/2\rho$ to achieve $\rho$-zCDP. Moreover, we have

$$\widehat{\theta}^T e_1 = -\frac{n}{2} - \frac{(1 - 2\alpha)n}{2}v_1 + \mathcal{N}(0, \sigma^2).$$

Thus, the mis-classification error is

$$\mathbb{P}[\widehat{\theta}e_1 > 0] = \Phi\left( \frac{n\left[1 - (1 - 2\alpha)v_1\right]}{2\sigma} \right) \leq e^{-\frac{n^2(1 - \beta + 2\alpha\beta)^2\rho}{4G^2}}.$$

As a result, the sample complexity to achieve $1 - \gamma$ accuracy is

$$n = O\left( \sqrt{\frac{4G^2 \log \frac{1}{\delta}}{(1 - \beta + 2\beta\alpha)^2 \cdot \rho}} \right)$$

The sensitivity $G = \sqrt{1 + \epsilon^2 p}$ here is dimension-dependent.

### C.2    RANDOM PERTURBATION

Now we consider the random perturbation. Denote $\{v_i\}_{i=1}^n \subseteq \mathbb{R}^p$ a sequence of i.i.d. copies of a random vector $v$. Consider the binary classification problem with training set $\{(x_i, y_i)\}_{i=1}^n$. Here $x_i = e_1 + v_i$ if $y_i = 1$ and $x_i = -e_1 + v_i$ if $y_i = -1$. Then, the loss function is $\mathcal{L}(\theta) = \frac{1}{n}\sum_{i=1}^n \log\left(1 + e^{-y_i\theta^T x_i}\right)$. The one-step iterate of DP-GD from 0 outputs

$$\widehat{\theta} = -\eta \sum_{i=1}^n (-y_i x_i) + \mathcal{N}(0, \sigma^2 I_p)$$

with $\sigma^2 = G^2/2\rho$ and $G = \sup_{v_i} \sqrt{1 + \|v_i\|^2}$ Assume that $v_i$ is symmetric, that is $y_i v_i$ has the same distribution as $-y_i v_i$. Then, it holds

$$\sum_{i=1}^n y_i x_i = ne_1 + \sum_{i=1}^n v_i =: \mu_n.$$

The mis-classification error is now given by

$$\mathbb{P}[\widehat{\theta}^T e_1 < 0] = \mathbb{P}[\mathcal{N}(\mu_n^T e_1, \sigma^2) < 0].$$

Assume that $\|v_i\|_\infty = \epsilon < 1$. Then, we have $\mu_n^T e_1 \geq n - \epsilon n$ and the sample complexity is $O\left( \sqrt{\frac{4G^2 \log(1/\delta)}{(1 - \epsilon)^2 \rho}} \right)$. with $G = \sqrt{1 + \epsilon^2 p}$.

# D  REMEDY TO NON-ROBUSTNESS

## D.1  DETAILS OF THE NORMALIZATION

Consider the case where the feature is shifted by a constant offset $v$. The feature of the $k$-th class is

$$\widetilde{x}_i = x_i - \frac{1}{n}\sum_{i=1}^{n} x_i = \widetilde{M}_k = M_k + v$$

with $M_k$ being the $k$-th column of the ETF $M$.

The offset $v$ can be canceled out by considering the differences between the features. That is, we train with the feature $\widetilde{M}_k - \frac{1}{K}\sum_{j=1}^{K}\widetilde{M}_j$ for the $k$-th class. In fact, let $P_k$ be the $k$-th column of $P$ and we have

$$
\begin{aligned}
\widetilde{M}_k - \frac{1}{K}\sum_{j=1}^{K}\widetilde{M}_j &= M_k - \frac{1}{K}\sum_{j=1}^{K}M_j \\
&= \sqrt{\frac{K}{K-1}}\left[\left(P_k - \frac{1}{K}\sum_{i=1}^{K}P_i\right) - \frac{1}{K}\sum_{j=1}^{K}\left(P_j - \frac{1}{K}\sum_{i=1}^{K}P_i\right)\right] \\
&= \sqrt{\frac{K}{K-1}}\left(P_k - \frac{1}{K}\sum_{j=1}^{K}P_j\right) = M_k.
\end{aligned}
$$

## D.2  PROOF OF THEOREM 5

*Proof of Theorem 5.*  Consider the case with $K = 2$ and a projection vector $\widehat{P} = (e_1 + \Delta)$ with some perturbation $\Delta = (\Delta_1, \cdots, \Delta_p)$ such that $\|\Delta\|_\infty \le \beta_0$ for some $0 < \beta_0 \ll p$. $\widehat{P}$ can be generated by the pre-training dataset or the testing dataset. Consider training with features $\widetilde{x}_i = \widehat{P}x_i$. Then, the sensitivity of the NoisyGD is $G = \sup_v |\widehat{P}^T(e_1 + v)| = 1 + \beta + \beta|\Delta_1| + \beta(\sum_{j=1}^{p}|\Delta_j|) \le 1 + \beta(1 + \beta_0 + p\beta_0)$. The output of Noisy-GD is then given by

$$\widehat{\theta} = -\widehat{P}\cdot\left(\sum_{i=1}^{n} y_i\widetilde{x}_i\right) + \mathcal{N}(0, \sigma^2).$$

Moreover, for any testing data point $e_1 + v$, define

$$\widehat{\mu}_n = -\left(\sum_{i=1}^{n} y_i\widetilde{x}_i\right)\widehat{P}^T(e_1 + v) = (e_1 + V)^T\widehat{P}\widehat{P}^T(e_1 + v)$$

with $V = \frac{1}{n}\sum_{i=1}^{n} v_i =: (V_1, \cdots, V_p)$.

We now divide $\widehat{\mu}_n$ into four terms and bound each term separately.

For the first term $e_1^T\widehat{P}\widehat{P}^T e_1$, it holds

$$e_1^T\widehat{P}\widehat{P}^T e_1 = (1 + e_1^T\Delta_1)^2 \le (1 - \beta_0)^2.$$

For the second term $V^T\widehat{P}\widehat{P}^T e_1$, we have

$$V^T\widehat{P}\widehat{P}^T e_1 = V_1 + V^T\Delta$$

Note that $V_1$ is the average of $n$ i.i.d. random variables bounded by $\beta$. By Hoeffding's inequality, we obtain

$$|V_1| \le \frac{\beta\log\frac{2}{\gamma}}{\sqrt{n}}, \text{ with probability at least } 1 - \gamma.$$

Similarly, with confidence $1 - \gamma$, it holds

$$|V^T \Delta| \leq \frac{p \beta \beta_0 \log \frac{2}{\gamma}}{\sqrt{n}}.$$

The third term $e_1^T \widehat{P} \widehat{P}^T v$ can be bounded as

$$|e_1^T \widehat{P} \widehat{P}^T v| = (1 + \Delta_1) \left( \sum_{j=1}^{p} v_i (1 + \Delta_i) \right) \leq (1 + \beta_0) \left( \beta + \beta_0 \sqrt{p \log \frac{2}{\gamma}} \right),$$

where the last inequality is a result of the Hoeffding's inequality by assuming that each coordinate of $v$ are independent of each others. Moreover, without further assumptions on the independence of each coordinate of $v$, we have

$$|e_1^T \widehat{P} \widehat{P}^T v| = (1 + \Delta_1) \left( \sum_{j=1}^{p} v_i (1 + \Delta_i) \right) \leq (1 + \beta_0)(\beta + \beta_0 p).$$

Using the Hoeffding's inequality again, for the last term $V^T \widehat{P} \widehat{P}^T (e_1 + v)$, it holds

$$|V^T \widehat{P} \widehat{P}^T (e_1 + v)| \leq \frac{(\beta + \beta_0 \sqrt{p})(1 + \beta + \beta_0 + \beta \beta_0 \sqrt{p}) \log \frac{4}{\gamma}}{\sqrt{n}}$$

with confidence $1 - \gamma$ if we assume that all coordinates of $v$ are independent of each other. Without further assumptions on the independence of each coordinate of $v$, we have

$$|V^T \widehat{P} \widehat{P}^T (e_1 + v)| \leq \frac{(\beta + \beta_0 p)(1 + \beta + \beta_0 + \beta \beta_0 p) \log \frac{2}{\gamma}}{\sqrt{n}}.$$

$\square$

# E  SOME CALCULATIONS ON RANDOM INITIALIZATION

## E.1  GAUSSIAN INITIALIZATION WITHOUT OFFSET

For Gaussian initialization $\xi = (\xi_1, \cdots, \xi_p) \sim \mathcal{N}(0, I_p)$, we have

$$\hat{\theta} = \xi - \eta \left( \frac{n}{2} \cdot \frac{-e^{-\xi_1}}{1 + e^{-\xi_1}} \cdot \left( - \begin{bmatrix} -1 \\ 0 \\ \vdots \\ 0 \end{bmatrix} \right) + \frac{n}{2} \cdot \frac{-e^{-\xi_1}}{1 + e^{-\xi_1}} \cdot \begin{bmatrix} 1 \\ 0 \\ \vdots \\ 0 \end{bmatrix} + \mathcal{N}(0, \frac{G^2}{2\rho} I_p) \right)$$

$$= \xi + \eta \left( \frac{e^{-\xi_1}}{1 + e^{-\xi_1}} \cdot \begin{bmatrix} n \\ 0 \\ \vdots \\ 0 \end{bmatrix} + \mathcal{N}(0, \frac{G^2}{2\rho} I_p) \right)$$

The sensitivity is $\frac{e^{-\xi_1}}{1 + e^{-\xi_1}} < 1$. Consider $x = (-1, 0, \cdots, 0)^T$. We have

$$\widehat{\theta}^T x = -\xi_1 + \eta \left( -\frac{n e^{-\xi_1}}{1 + e^{-\xi_1}} \right) + \mathcal{N}(0, \frac{G^2}{2\rho}) =: \mu_{\xi_1, n} + \mathcal{N}(0, \frac{G^2}{2\rho}).$$

The mis-classification error is

$$\mathbb{P}[\widehat{\theta}^T x > 0] = \mathbb{E}_{\xi_1 \sim \mathcal{N}(0,1)} \mathbb{P} \left[ \mathcal{N} \left( \mu_{\xi_1, n}, \frac{G^2}{2\rho} \right) > 0 \Big| \xi_1 \right]$$

$$= \mathbb{E}_{\xi_1 \sim \mathcal{N}(0,1)} \left[ \Phi \left( \frac{\sqrt{2\rho} \mu_{\xi_1, n}}{G} \right) \right]$$

### E.2 Gaussian Initialization with Off-Set

Denote $x_1 = -e_1 + v$ and $x_2 = e_1 + v$ with $\|v\|_\infty \leq \beta$. For the logistic loss $\ell(y, \theta^T x) = \log(1 + e^{-y\theta^T x})$, we have

$$g(\theta, y \cdot x) := \nabla_\theta \ell(y, \theta^T x) = \frac{e^{-y\theta^T x}}{1 + e^{-y\theta^T x}}(-yx).$$

Denote

$$g_1(\theta) = g(\theta, -1 \cdot x_1) = \frac{e^{\theta^T x_1}}{1 + e^{\theta^T x_1}} x_1$$

and

$$g_2(\theta) = g(\theta, 1 \cdot x_2) = \frac{e^{-\theta^T x_2}}{1 + e^{-\theta^T x_2}}(-x_2).$$

If we shift the feature by some vector $v$, then the loss function is

$$R_n = \frac{n}{2} \log(1 + e^{\theta^T x_1}) + \frac{n}{2} \log(1 + e^{-\theta^T x_2}).$$

And the gradient is

$$\nabla_\theta R_n(\theta) = \frac{n}{2} \left(g_1(\theta) + g_2(\theta)\right).$$

Thus, the output of one-step NoiseGD is given by

$$\widehat{\theta} = \theta_0 - \frac{\eta n}{2} \left[g_1(\theta_0) + g_2(\theta_0) + \mathcal{N}(0, \sigma^2)\right].$$

Let $\mu_\xi = \xi - \frac{\eta n}{2} \left[g_1(\xi) + g_2(\xi)\right].$ Then, we have

$$\mu_\xi^T e_1 = \xi_1 - \frac{\eta n e^{\xi^T x_1}}{2 + 2e^{\xi^T x_1}}(-1 + v_1) + \frac{\eta n e^{\xi^T x_2}}{2 + 2e^{\xi^T x_2}}(1 + v_1).$$

And the mis-classification error is

$$\mathbb{E}_\xi \left(\Phi\left(-\frac{\sqrt{2\rho}\mu_\xi^T e_1}{G}\right)\right).$$

