# OpenReview forum: "Neural Collapse meets Differential Privacy:  Curious behaviors of NoisySGD with Near-Perfect Representation Learning"
_ICLR.cc/2024/Conference — Submitted to ICLR 2024_

### Official Review · Reviewer_dsyA · 2023-10-31

**Soundness:** 1 poor
**Presentation:** 3 good
**Contribution:** 1 poor
**Rating:** 3
**Confidence:** 5

**Summary:**

Recent study by De et al. (2022) reports that large-scale representation learning via pre-training on a gigantic dataset significantly enhances differentially private learning in downstream tasks. While the exact behaviors of NoisySGD on these problems remain intractable to analyze
theoretically, the authors consider an idealized setting of a layer-peeled model for representation learning by neural collapse.

The writing is good and the results seem interesting, which attracts me to check their proof. The proofs are very simple. $M_k,k=1,\cdots,K$ form an ETF frame, which separate categories very well, the zero initialization makes $f_0(M_k)=0$ for all $k=1,\cdots,K$, which is very weird. The one-step NoisyGD seems not useful at all.

 I am not sure your results are for NoisySGD or NoisyGD. In introduction section, your statements are all about NoisySGD, but the other parts for NoisyGD. Moreover, there is no definition about NoisySGD at all in the whole paper.

**Strengths:**

The presentation is good.

**Weaknesses:**

The results are not meaningful.

**Questions:**

How about $f_W(x)=Wx+b$?

---

> ### Author Response · Authors · 2023-11-17
>
> It appears that our main contribution may have been overlooked by the reviewer. We would like to highlight it once more and also address the reviewer's question as follows.
>
>
>
>
> $\mathbf{Comments:}$ $M_k, k = 1,\cdots, K$ form an ETF frame, orm an ETF frame, which separate categories very well, the zero initialization makes $f_0(M_k) = 0$
> , which is very weird.
>
> $\mathbf{Response:}$ We would like to clarify that $0$ initialization is ubiquitous in training SGD or DP-SGD and we are not sure which part is weird.
> Even though, $f_0(M_k) = 0$ for all $k$, the training process will push the parameter to fit the the separate categories after iterations.
>
> $\mathbf{Comments:}$ The one-step NoisyGD seems not useful at all.
>
> $\mathbf{Response:}$
> For perfect collapse, the fact that 1-iteration, full-batch GD suffices, and the fact that it is free of hyperparameters (any learning rate, any class imbalance, any dimension, any meaningful loss function)  come at a surprise in a sense that — even such a much weaker and non-adaptive algorithm works.
>
>  In more realistic, imperfect collapse scenarios, our theory can be extended to multiple iterations. However, it is obvious that dimension dependency can not be mitigated using multi-iterations. Therefore, it is not clear why one-iteration NoisyGD would be deemed without merit.
>
> $\mathbf{Comments:}$ I am not sure your results are for NoisySGD or NoisyGD. In introduction section, your statements are all about NoisySGD, but the other parts for NoisyGD. Moreover, there is no definition about NoisySGD at all in the whole paper.
>
> $\mathbf{Response:}$ Our theoretical framework specifically addresses NoisyGD. In our experimental analysis, however, we assess the performance of both NoisyGD and NoisySGD. We appreciate the reviewer highlighting the omission of NoisySGD's definition, and we have now included it in Section 2.2.
>
>
>
>
>
> $\mathbf{Comments:}$ The proofs are very simple.
>
> $\mathbf {Response:}$
> We agree that our analysis is simple, but in light of Point 1,2,3 in the official rebuttal above, we believe the simplicity of the analysis is a feature rather than a bug! After all, we believe the reviewer agree that our results are interesting and new.
> To reiterate, we would like to highlight once more the contributions and novel findings of our study.
> Isn’t it better to have a simpler proof of a fundamental result than a highly technical (and harder to verify) proof?
>
>
> 1. It appears that our crucial observation--- that DP fine-tuning is non-robust to perturbations---was not acknowledged by the reviewer, a point we emphasized in Sections 3.2 and 3.3.
> The “Approximate Neural Collapse” case shows that the remarkable property is quite fragile if we train with standard DP-GD or DP-SGD.  This actually led to various actionable algorithmic modifications to these private learning methods that make these methods substantially more robust — in theory and in practice!
>
> 2.  In the perfect neural collapse case, our analysis also works perfectly for DP-SGD and it works for multiple iterations. As we stated above, many interesting facts come at a surprise in a sense that — even such a much weaker and non-adaptive algorithm works.
>
> 3. Existing analysis of DP-GD and DP-SGD focuses on suboptimality in the surrogate losses.  The sample complexity is polynomial  $d/\gamma$  or $d/ \sqrt{\gamma}$ (in the strongly convex case) to achieve $(1-\gamma)$-accuracy.  We actually directly studied the ``0-1" loss!  And we showed that the sample complexity is  $\log(1/\gamma)$.       This is an exponential improvement and shows an exponential benefit of strong representation learning.
>
> $\mathbf{Questions:}$ How about $f_W(x) = Wx + b$?
>
> $\mathbf {Response:}$ In the case of perfect collapse, incorporating an offset term $b$  is unnecessary. For binary classification problem with $\alpha$-class imbalance, straightforward computation reveals that the output of NoisyGD is given by $(n e_1,(1 - 2\alpha)) + \mathcal{N}(0,\sigma^2 I)$ where the $1 - 2\alpha$ component arises from the offset $b$ and the direction associated with $b$ is simply Gaussian noise. This noise could potentially detract from performance.
>
> For a non-perfect collapse scenario, a similar computation indicates that an offset $b$ does not eliminate the perturbation. Consequently, the outcome retains its dimension-dependence.

---

> > ### Comment · Reviewer_dsyA · 2023-11-23
> > **Response to the authors**
> >
> > Dear authors,
> >
> > Thanks for your reply. I keep my score since the result may be not meaningful.

---

> > > ### Author Response · Authors · 2023-11-23
> > >
> > > Thank you for your feedback. Could the reviewer possibly offer further explanation as to why our theoretical and experimental observations regarding non-robustness may not be meaningful? Additionally, would you be able to suggest any other references that might provide more significant insights? Your opinions will be appreciated.

---

### Official Review · Reviewer_KJYE · 2023-11-10

**Soundness:** 2 fair
**Presentation:** 3 good
**Contribution:** 2 fair
**Rating:** 5
**Confidence:** 3

**Summary:**

This paper studies theoretical analysis for differentially private fine-tuning under neural collapse. Specifically, this paper shows that if the neural collapse is assumed, and we only fine-tune last layer, the accuracy bound is indepedent of dimension and only related to privacy parameter. If the neural collapse is not perfect on private dataset, this paper also shows that perturbation on the features, class imbalance would require the accuracy to be depedent on the dimension. This paper also propose data normalization and PCA to mitigate this non-robustness issue.

**Strengths:**

This paper provides first theoretical understanding of DP fine-tuning reduces the error rate on down-streaming task. The setting of neural collapse is interesting and may be enlightening for potential future research.

**Weaknesses:**

1. Typos: In theorem 2, is it $\gamma$ accuracy or $1-\gamma$ accuracy?
2. All of the proofs only analyze one-step Noisy-GD algorithm under a very strong neural collapse assumption. This setting is too simple and might not be reflecting what is happening in De at al (2022). If perfect neural collapse holds, then there is no need for further training. For example,  in theorem 2, you can set the clipping threshold G to be very small (near zero) to get near zero error rate. This suggests that you don't have to train on the private data if the neural collapse is assumed. This bound might not be very useful.
3. The proposed tricks are not demonstrated empirically on real datasets.
4. The proof is simple and the technical contribution is limited.

**Questions:**

1. Is there any empirical improvement by using the proposed data normalization and PCA tricks? I am curious because DP-PCA would also needs privacy-utility trade-off that needs to be accounted.

---

> ### Author Response · Authors · 2023-11-17
>
> Thank you for your valuable comments. It appears there may have been some misunderstandings in the review, potentially due to our failure to emphasize certain aspects within our original manuscript. We would like to address these points in the following response
>
>
> $\mathbf{Weakness 1:}$ Typos: In theorem 2, is it $\gamma$ accuracy or $1 - \gamma$ accuracy?
>
> $\mathbf{Response:}$ Thank you for pointing it out. It is $1 - \gamma$ accuracy and we have corrected it.
>
> $\mathbf{Weakness 2:}$ All of the proofs only analyze one-step ... might not be very useful. &
> $\mathbf{Weakness 4:}$ The proof is simple and the technical contribution is limited.
>
> $\mathbf{Response:}$ We have emphasized our main contributions in the official comments in details and we would like to emphasize them again here shortly.
>
> 1. In addition to examining the ideal scenario of perfect neural collapse, our research also explores the concept of approximate collapse, which is more prevalent in practical situations. Our results suggest that the extraordinary dimension-independent properties observed during perfect neural collapse is inherently fragile. This non-robustness is also evident in our empirical studies for both NoisyGD and NoisySGD using real datasets!
>
> 2. We wish to clarify for the reviewer that the example where the $G$ is near zero, as mentioned by the reviewer, does not align with our theory. In fact, in the proof, we choose $G$ to be the sensitivity of the gradient, and our theory is valid only when $G\geq 1$ for perfect collapse, or when $G \geq \sqrt{1 + \beta^2 p}$ for imperfect collapse.  It is worth noting that even if we clip the data point with a small $G$, this does not alter our results. In fact, in the case where $K=2$, as detailed in the proof of Theorem 3 (similar results hold for the more general case, Theorem 2, and the imperfect case in Section 3.2 and Section 3.3) in Appendix A.2, $n/2$ is scaled to $Gn/2$ if $G<1$, and a smaller $G$ does not imply a smaller mis-classification error.
>
> 3. Existing analysis of DP-GD and DP-SGD focuses on suboptimality in the surrogate losses.  The sample complexity is polynomial  $d/\gamma$  or $d/ \sqrt{\gamma}$ (in the strongly convex case) to achieve $(1-\gamma)$-accuracy.  We actually directly studied the ``0-1" loss!  And we showed that the sample complexity is  $\log(1/\gamma)$. This is an exponential improvement and shows an exponential benefit of strong representation learning.
>
> 4. We agree that our analysis is simple, but in light of those points in this response and the official comments above, we believe the simplicity of the analysis is a feature rather than a bug! After all, we believe the reviewer agree that our results are interesting and new.  Isn’t it better to have a simpler proof of a fundamental result than a highly technical (and harder to verify) proof?
>
> $\mathbf{Weakness 3:}$ The proposed tricks are not demonstrated empirically on real datasets.
>
> $\mathbf{Response:}$ We wish to clarify for the reviewer that our investigation includes empirical evidence of the non-robustness of DP-finetuning to perturbations using real datasets. Specifically, our Figure 2(b), which uses a real dataset, demonstrates that SGD without DP is robust to increasing dimensions. Conversely, the accuracy of DP-SGD decreases significantly with rising dimensions.
>
> Additionally, the phenomenon of neural collapse is a widely observed phenomenon in deep learning, as evidenced by various references cited in our paper, including the study available at https://www.pnas.org/doi/10.1073/pnas.2103091118. Related to transfer learning, a notable reference is a recent paper presented at NeurIPS, which provides evidence of neural collapse during the fine-tuning process (https://openreview.net/pdf?id=xQOHOpe1Fv).
>
> $\mathbf{Questions:}$ Is there any ... to be accounted.
>
> $\mathbf{Response:}$ DP-PCA has been widely adopted in DP-SGD to enhance empirical performance, as evident in ``Deep Learning with Differential Privacy" (https://arxiv.org/abs/1607.00133). We provide a perspective from neural collapse theory to explain the efficacy of PCA.
> Due to the time constraints of the rebuttal period, we have not conducted experiments using DP-PCA at this stage.
>
> The reviewer is correct that DP-PCA requires a privacy-utility trade-off. However, the impact of the privacy budget on PCA's accuracy is not the focus of our paper.
> It's important to note that the recent theory on DP-PCA, such as the one described in ``https://arxiv.org/abs/2205.13709", cannot be directly applied to our setting since their accuracy is measured by the angle between eigenvectors.
> In contrast, our investigation into the 0-1 loss highlights the importance of the infinity norm between eigenvectors.
> Therefore, it would be interesting to establish theoretical guarantees for DP-PCA under the infinity norm.

---

### Official Review · Reviewer_MQQr · 2023-11-10

**Soundness:** 3 good
**Presentation:** 3 good
**Contribution:** 2 fair
**Rating:** 3
**Confidence:** 3

**Summary:**

The authors use neural collapse theory to analyze the behavior of last-layer fine-tuning with DP. They show that dimension independence emerges in a certain sense under perfect neural collapse and that small perturbations in the train and test data can disturb this independence. They show that data normalization and dimensions reduction can recover the dimension independence in the face of such perturbations.

**Strengths:**

1. The general phenomenon explored in this paper (i.e. the empirical success of DP deep fine-tuning in high dimensions) is very interesting and timely.
2. The results are presented with a high degree of technical precision and fluency.
3. The result of Theorem 2 seems surprising and interesting to me, although I don't yet have a strong intuitive understanding of the proof.

**Weaknesses:**

1. **It may be difficult for some readers to understand:** Given the topic, I imagine many readers will be familiar with differential privacy but less familiar with neural collapse. As a result, the third paragraph of the introduction and the corresponding figure 1 will be meaningless to them without more explanation. Some of the introductory material is present in section 2.2, but it is a bit technical and not well-suited to newcomers. I would recommend giving a high level explanation of neural collapse in the introduction to help readers.

**Questions:**

1. Introduction, paragraph 1, last line: do you mean "no-privacy utility tradeoff"
2. Bottom of page 7, Sigma is missing a backslash.
3. The dimension independence of Theorem 5 requires $\beta_0$ to scale with $p$, but the non-robustness results in 3.2 and 3.3 would also become dimension independent if $\beta$ chosen to vary with $p$ in a similar way. Because of this, it's not clear what we are gaining from dimension reduction in section 4.2.
2. It's not clear to me how this analysis of neural collapse applies to full fine tuning. The success of DP full fine tuning is most surprising because of the many total parameters of the networks (not just in the last layer). Neural collapse may explain some of the dynamics of last layer fine-tuning as presented in this paper, but clearly something interesting must be happening at intermediate layers in the full fine-tuning setting. I think it would be very helpful to mention whether there is any way that these results might shed light on the dynamics of intermediate layers.

---

> ### Author Response · Authors · 2023-11-17
>
> We are grateful for your valuable feedback. In particular, Question 4 has provided us with insightful perspectives. We would like to offer the following response in return.
>
> $\mathbf{Weakness:}$ It may be difficult for some readers to understand... I would recommend giving a high level explanation of neural collapse in the introduction to help readers.
>
> $\mathbf{Response:}$ Thank you for these comments. We have added a high-level explanation of the neural collapse and the $K$-simplex equiangular tight frame (ETF) to the manuscript in the introduction. Neural collapse is the phenomenon where features corresponding to different classes are separable enough. An illustrative example for the case where $K=2$ is as follows: an ETF $M$ can be represented as $M = [-m_1, m_1]$ for some vector $m_1 \in \mathbb{R}^{p}$, where the features corresponding to the label $y=1$ are $m_1$, and the features corresponding to the label $y=-1$ are $-m_1$.
>
> $\mathbf{Question 1:}$ Introduction, paragraph 1, last line: do you mean "no-privacy utility tradeoff".
>
> $\mathbf{Response:}$ Thank you for pointing out this typo. We have corrected it.
>
> $\mathbf{Question 2:}$ Bottom of page 7, Sigma is missing a backslash.
>
> $\mathbf{Response:}$ Corrected.
>
> $\mathbf{Question 3:}$ The dimension independence of Theorem 5 requires $\beta_0$ to scale with $p$, but the non-robustness results in 3.2 and 3.3 would also become dimension independent if $\beta$ chosen to vary with $p$ in a similar way. Because of this, it's not clear what we are gaining from dimension reduction in section 4.2.
>
> $\mathbf{Response:}$ As detailed in Section 4.3, for PCA we have established that $\beta_0 \leq \frac{1}{\sqrt{m}}$ with high probability  where $m$ represents the sample size used in PCA, effectively scaling $p$ to $p/\sqrt{m}.$
> Our theoretical framework, at this stage, does not encompass DP-PCA, which is actually implemented in DP-SGD. Should DP-PCA be taken into account,
> $\beta_{0}$ would be influenced by the privacy budget. While our current paper does not concentrate on this aspect, understanding the convergence of the eigenvector in DP-PCA, particularly in terms of the infinity norm, necessitates a separate and thorough analysis.
>
> $\mathbf {Question 4:}$
> It's not clear to me how this analysis of neural collapse applies to full fine tuning ... these results might shed light on the dynamics of intermediate layers.
>
> $\mathbf{Response:}$ Thank you for drawing attention to this intriguing topic. Our empirical findings suggest that, except in the final layer, neural collapse is typically not observed—even in settings without Differential Privacy. This is also corroborated by another paper "Exploring Deep Neural Networks via Layer-Peeled Model: Minority Collapse in Imbalanced Training" (https://arxiv.org/pdf/2101.12699.pdf), which explains that, from a mathematical standpoint, an Equiangular Tight Frame (ETF) is the optimal solution for Empirical Risk Minimization (ERM) when considering the features of the last layer.
>
> Recent research, such as "The Tunnel Effect: Building Data Representations in Deep Neural Networks" by Masarczyk et al., presented at NeurIPS 2023, has made some novel observations. These include the discovery that the last few layers of deep neural networks exhibit low-rank structures, and their performance is resemblance to that of linear models. Within this context, neural collapse in the final layer can be seen as a special case of these low-rank structures, which are also present in transfer learning scenarios.
>
> Given these developments, it would be a valuable line of inquiry for our future research to examine whether fine-tuning the last few layers with NoisySGD could reveal low-rank structures, like neural collapse --- or perhaps a partially neural collapse influenced by the noise. Due to time limitations, we will defer this investigation to subsequent studies.

---

### Author Response · Authors · 2023-11-17
**Official Rebuttal to Emphasize Our Novelty and Contributions**

We are grateful for the thoughtful feedback provided by all reviewers. However, it has come to our attention that the full extent of our work's primary contributions and innovations, especially our significant finding regarding the non-robustness of Differential Privacy (DP) in fine-tuning, substantiated by both theoretical framework (using approximate neural collapse) and empirical evidence using a real dataset, was not fully acknowledged by most reviewers.  In these official comments, we would like to emphasize the central contributions and the novel elements of our research.

1.  In the perfect neural collapse case, our analysis also works perfectly for DP-SGD and it works for multiple iterations.   The fact that 1-iteration, full-batch GD suffices, and the fact that it is free of hyperparameters (any learning rate, any class imbalance, any dimension, any meaningful loss function)  comes at a surprise in a sense that — even such a much weaker and non-adaptive algorithm works.

2. Existing analysis of DP-GD and DP-SGD focuses on suboptimality in the surrogate losses.  The sample complexity is polynomial  $d/\gamma$  or $d/ \sqrt{\gamma}$ (in the strongly convex case) to achieve $(1-\gamma)$-accuracy.  We actually directly studied the ``0-1" loss!  And we showed that the sample complexity is  $\log(1/\gamma)$.       This is an exponential improvement and shows an exponential benefit of strong representation learning.

3. We also analyzed the “Approximate Neural Collapse” case in Section 3.2 and 3.3 which is more realistic and shows that the remarkable property is quite fragile if we train with standard DP-GD or DP-SGD.  This actually led to various actionable algorithmic modifications to these private learning methods that make these methods substantially more robust — in theory and in practice!

4. We agree that our analysis is simple, but in light of Point 1,2,3 above, we believe the simplicity of the analysis is a feature rather than a bug! After all, we believe the reviewers and AC agree that our results are interesting and new.  Isn’t it better to have a simpler proof of a fundamental result than a highly technical (and harder to verify) proof?

---

### Meta-Review · Area_Chair_jNHb · 2023-12-11

**Metareview:**

(a) Summarize the scientific claims and findings of the paper based on your own reading and characterizations from the reviewers.

This manuscript attempts to understand and explain the dimension dependence of high dimensional model training with DPSGD. The lens the authors choose to use is neural collapse: in typical training of classifiers, the last layer representation converges to their respective class centers, and those centers are mutually orthogonal. Theoretically, it is demonstrated that neural collapse implies dimension independence for DPSGD output. Under minor perturbations, the performance becomes dependent on the dimension, which implies lack of robustness in DPSGD. Empirically, it is shown that imagenet pretrained models do not suffer from high dimensions when finetuned on CIFAR10. However, a perturbation to the representations result in degraded utility-privacy tradeoff as the authors predicted. Under fixed perturbation, it is experimentally demonstrated that PCA and data normalization effectively makes it robust, eliminating the dimension dependence issue. Under random perturbation, it is theoretically demonstrated that PCA effectively mitigates dimension dependence.

(b) What are the strengths of the paper?

The motivation of this paper on bringing the recent advances in neural collapse and the interpretation of the training dynamics of neural networks to differential private model training is interesting. It is possible that there is an exciting new observations and research directions at this intersection. Such investigations sometimes lead to exciting new results. In the case of this paper, it starts out ambitiously to bridge the gap between these two research areas, and find something new and significant. However, the results and observations made in the paper are straightforward and not insightful.

(c) What are the weaknesses of the paper? What might be missing in the submission?

Theoretically, most of the observations follow in a straightforward manner. For example, it is not surprising that a K-way classification does not have dimension dependence, since noise in any other directions are irrelevant. Similarly, if the perturbation is measured by L-infinity bound, then it is again not surprising that such perturbation will incur dimension dependence. The size of the perturbation is increasing with the dimension. All of the proofs only analyze one-step Noisy-GD algorithm under a very strong neural collapse assumption. This setting is too simple and might not be reflecting what is happening in De at al (2022). If perfect neural collapse holds, then there is no need for further training. For example, in theorem 2, you can set the clipping threshold G to be very small (near zero) to get near zero error rate. This suggests that you don't have to train on the private data if the neural collapse is assumed. This bound might not be very useful. Empirically, the experiments do not show how the proposed method of PCA and normalization can benefit the final performance.

**Justification For Why Not Higher Score:**

The positioning of this paper is not helping. Empirically, the authors did not observe any meaningfully interesting new results. It is fair to say that calling this paper realistic is a stretch. We are left to focus on the theoretical impact and contributions of this paper. Many of the results are straightforward once some version of neural collapse is assumed. Because of these inherent issues, the paper does provide any new insights into private training.

**Justification For Why Not Lower Score:**

N/A

---

### Decision · Program_Chairs · 2024-01-16

Reject